# Development of an efficient Sanger sequencing-based assay for detecting SARS-CoV-2 spike mutations

Ho Jae Lim[1,2☯], Min Young Park[1☯], Hye Soo Jung[1], Youngjin Kwon[1], Inhee Kim[1], Dong Kwan Kim[1], Nae Yu[1], Nackmoon Sung[3], Sun-Hwa Lee[1], Jung Eun Park[2]*, Yong-Jin Yang[1]*

**1** Department of Molecular Diagnostics, Seegene Medical Foundation, Seoul, Republic of Korea, **2** Department of Integrative Biological Sciences & BK21 FOUR Educational Research Group for Age-associated Disorder Control Technology, Chosun University, Gwangju, Republic of Korea, **3** Clinical Research Institute, Seegene Medical Foundation, Seoul, Republic of Korea

☯ These authors contributed equally to this work.
* yjyang@mf.seegene.com (YJY); jepark@Chosun.ac.kr (JEP)

## Abstract

Novel strains of severe acute respiratory syndrome coronavirus 2 (SARS-CoV-2) harboring nucleotide changes (mutations) in the spike gene have emerged and are spreading rapidly. These mutations are associated with SARS-CoV-2 transmissibility, virulence, or resistance to some neutralizing antibodies. Thus, the accurate detection of spike mutants is crucial for controlling SARS-CoV-2 transmission and identifying neutralizing antibody-resistance caused by amino acid changes in the receptor-binding domain. Here, we developed five SARS-CoV-2 spike gene primer pairs (5-SSG primer assay; 69S, 144S, 417S, 484S, and 570S) and verified their ability to detect nine key spike mutations (ΔH69/V70, T95I, G142D, ΔY144, K417T/N, L452R, E484K/Q, N501Y, and H655Y) using a Sanger sequencing-based assay. The 5-SSG primer assay showed 100% specificity and a conservative limit of detection with a median tissue culture infective dose ($TCID_{50}$) values of $1.4 \times 10^2$ $TCID_{50}$/mL. The accuracy of the 5-SSG primer assay was confirmed by next generation sequencing. The results of these two approaches showed 100% consistency. Taken together, the ability of the 5-SSG primer assay to accurately detect key SARS-CoV-2 spike mutants is reliable. Thus, it is a useful tool for detecting SARS-CoV-2 spike gene mutants in a clinical setting, thereby helping to improve the management of patients with COVID-19.

## Introduction

The World Health Organization (WHO) declared the coronavirus disease (COVID-19), which is caused by severe acute respiratory syndrome coronavirus 2 (SARS-CoV-2), as a global pandemic on March 11, 2020 [1]. SARS-CoV-2 is a highly transmissible virus and has a long incubation time before the manifestation of symptoms, such as fever, cough, shortness of breath, and diarrhea [2]. SARS-CoV-2 has a single-stranded, positive-sense RNA genome of

**Data Availability Statement:** All relevant data are within the paper and its Supporting Information files.

**Funding:** The authors received no specific funding for this work.

**Competing interests:** The authors have declared that no competing interests exist.

approximately 29.9 kb, which encodes several proteins, including the structural proteins, and spike (S) [3,4]. The S gene encodes the S1 and S2 subunits, and the S1 subunit contains an N-terminal domain and a receptor-binding domain, the latter of which is associated with human infections [5]. Across its genome, the virus accumulates mutations that are associated with its transmissibility, virulence, or resistance to some neutralizing antibodies [6,7].

SARS-CoV-2 variants have been recently identified, raising concerns abouta subsequent wave of the pandemic. Since the emergence of multiple variants, the WHO and the Centers for Disease Control and Prevention (CDC) have set up a classification scheme for monitoring the potential impact of emerging variants. The variants are classified into variants of interest (VOIs), variants of concern (VOCs), and variants of high consequence (VOHCs) [8,9]. Currently, VOIs, including eta (B.1.525), iota (B.1.526), kappa (B.1.617.1), unlabeled (B.1.617.3), and epsilon (B.1.427 and B.1.429) [10], have been associated with changes in receptor binding, reduced neutralization by antibodies generated against previous infection or vaccination, reduced efficacy of treatments, potential diagnostic effects, or a predicted increase in transmissibility or disease severity. VOCs, including alpha (B.1.1.7), beta (B.1.351), delta (B.1.617.2), and gamma (P.1) [10], have been associated with an increase in transmissibility and disease severity, a significant reduction in neutralization by antibodies generated during previous infection or vaccination, reduced effectiveness of treatments or vaccines, or diagnostic detection failures. The D614G substitution is the most prevalent mutation observed [11,12]. However, VOHC-related lineages have not yet been classified [10].

These lineages have been analyzed using next generation sequencing (NGS) methods and classified using Phylogenetic Assignment of Named Global Outbreak (PANGO) lineages [13]. The use of NGS has been very useful for obtaining accurate information on genetic variability and transmission [14]. However, as outbreaks occur sporadically and cannot be predicted, it is not always possible to have all resources required to perform the tests necessary to detect SARS-CoV-2 variants, especially in resource-limited settings [15]. To overcome this limitation, both PCR and Sanger sequencing have been applied [16,17].

Here, we aimed to accurately and rapidly detect nine key S mutations (ΔH69/V70, T95I, G142D, ΔY144, K417T/N, L452R, E484K/Q, N501Y, and H655Y) in strains classified as VOCs and/or VOIs using our laboratory-developed five SARS-CoV-2 S gene (5-SSG) primers via PCR assay in conjunction with Sanger sequencing. In addition, we compared the results of our assay with those of a commercially available NGS assay to evaluate its accuracy and reliability in detecting and identifying variants.

## Materials and methods

### Primer design

The 5-SSG primer assay was designed based on the Wuhan-CoV reference sequence (Wuhan-Hu-1, NCBI accession number NC_045512.2) [18,19]. Primers were modified from the Global Initiative on Sharing Avian Influenza Data (GISAID) database, with a frequency cut-off > 1%, applied with degenerative or inosine to optimize the melting temperature (Tm), avoid repetitive sequences, and include GC content > 65%, using Gene Runner (ver. 6.0) [20,21]. NCBI-Basic Local Alignment Search Tool (NCBI-BLAST) was used to optimize the specificity for SARS-CoV-2 [22]. Sequences were also screened based on alignments using the GISAID database for species selectivity. After these assessments, five targets were selected for validation using the S mutant assay, and M13 universal sequence primers were tagged for Sanger sequencing. The sequences of the 5-SSG primers (69S forward primer, `TGTAAAACGACGGCCAGTATTACCCTGACAAAGTTTTCAGATC`; 69S reverse primer, `CAGGAAACAGCTATGACGCGTTATTAACAATAAGTAGGGAC`; 144S forward primer, `TGTAAAACGACGGCCAGTCCACTGAG`

AAGTYTAACATAAT AAGAG; 144S reverse primer, CAGGAAACAGCTATGACTCACCAGGAG TCAAATA ACTTCTAT; 417S forward primer, TGTAAAACGACGGCCAGTGCTTTAGAACCA TT GGTAGATTTG; 417S reverse primer, CAGGAAACAGCTATGACGTTTGAGATTAG ACTT CCTAAACAATC; 484S forward primer, TGTAAAACGACGGCCAGTTCTAAYA AICTTGATT CTAAGGTTG; 484S reverse primer, CAGGAAACAGCTATGACCKCCT GTGCCTGTTAAACC ATT; 570S forward primer, TGTAAAACGACGGCCAGTGAAC TTCTACATGCACCAGCAAC; 570S reverse primer, CAGGAAACAGCTATGACCTG CATTCAGTTGAATCACCAC) are presented in Table 1. The 5-SSG primer-specific target mutants and lineages are summarized in S1 Table.

## Strain information and cultivation

To determine analytical specificity, 67 strains of viruses, bacteria, and fungi were used with or without respiratory pathogens, including 42 strains of virus (24 strains of SARS-CoV-2, Coronavirus OC43 and 229E, and 18 other viruses), 19 strains of bacteria, and six strains of fungi (Tables 2 and S2). The powdered nucleic acid of all strains used in this study was obtained from the following suppliers: Twist Bioscience (San Francisco, CA, USA), National Culture Collection for Pathogens (NCCP; Cheongju, Republic of Korea), American Type Culture Collection (ATCC; Manassas, VA, USA), Zeptometrix (Buffalo, NY, USA), Korea Bank for Pathogen Viruses (KBPV; Seoul, Republic of Korea), National Institute for Biological Standards and Control (NIBSC; Potters Bar, United Kingdom), Korean Collection for Type Cultures (KCTC; Jeongeup, Republic of Korea), and Korean Culture Center of Microorganisms (KCCM; Seoul, Republic of Korea). Other detailed strain information, including lineages and CDC classification, is shown in Table 2.

## Clinical specimen collection and storage

As part of the routine procedure using the Allplex™ SARS-CoV-2 assay for SARS-CoV-2 testing (Seegene Inc., Seoul, Republic of Korea), anonymized residual of 17 SARS-CoV-2 positive nasopharyngeal swab specimens of patients diagnosed with SARS-CoV-2 positive between February and June 2021 were obtained and used for this study. All samples were processed using an automated nucleic acid extraction system, namely MagNA Pure 96 (Roche, Basel, Switzerland), according to the manufacturer's protocol, and stored at −80˚C until use [23].

**Table 1. Oligonucleotide primers used for one step PCR to detect S mutations.**

| Primer | Type | Start | End | Sequences (5′-3′) | Tm (˚C) | Size (bp) |
|--------|------|-------|-----|-------------------|---------|-----------|
| 69S | F. primer | 21672 | 21696 | TGTAAAACGACGGCCAGT**ATTACCCTGACAAAGTTTTCAGATC** | 65.8 | 294 |
|     | R. primer | 21907 | 21930 | CAGGAAACAGCTATGAC**GCGTTATTAACAATAAGTAGGGAC** | 62.8 | |
| 144S | F. primer | 21843 | 21869 | TGTAAAACGACGGCCAGT**CCACTGAGAAGTYTAACATAATAAGAG** | 63.8 | 513 |
|      | R. primer | 22295 | 22320 | CAGGAAACAGCTATGAC**TCACCAGGAGTCAAATAACTTCTAT** | 64.1 | |
| 417S | F. primer | 22226 | 22249 | TGTAAAACGACGGCCAGT**GCTTTAGAACCATTGGTAGATTTG** | 65.2 | 759 |
|      | R. primer | 22923 | 22949 | CAGGAAACAGCTATGAC**GTTTGAGATTAGACTTCCTAAACAATC** | 65.1 | |
| 484S | F. primer | 22874 | 22898 | TGTAAAACGACGGCCAGT**TCTAAYAAICTTGATTCTAAGGTTG** | 62.8 | 375 |
|      | R. primer | 23192 | 23213 | CAGGAAACAGCTATGAC**CKCCTGTGCCTGTTAAACCATT** | 66.3 | |
| 570S | F. primer | 23108 | 23130 | TGTAAAACGACGGCCAGT**GAACTTCTACATGCACCAGCAAC** | 66.2 | 739 |
|      | R. primer | 23790 | 23811 | CAGGAAACAGCTATGAC**CTGCATTCAGTTGAATCACCAC** | 64.5 | |

Primers for specific target mutations in Wuhan-Hu-1-CoV were designed and conserved regions of the S gene are highlighted in bold. Primers were extended by tagging the 5′ side with M13 as a universal sequencing primer. Abbreviations: T$_m$, melting temperature; F. primer, forward primer; R. primer, reverse primer; Δ, deletion; Y, C or T; K, G or T; I, inosine.

**Table 2. PCR results and lineage information associated with the respiratory pathogens used in this study.**

| Group | Strain | Source | Lineage | CDC classification | PCR result |
|---|---|---|---|---|---|
| Virus | SARS-CoV-2 | Twistbio-601443 | alpha (B.1.1.7) | VOC | Positive |
| | SARS-CoV-2 | Twistbio-678597 | beta (B.1.351) | VOC | Positive |
| | SARS-CoV-2 | Twistbio-710528 | alpha (B.1.1.7) | VOC | Positive |
| | SARS-CoV-2 | Twistbio-79683 | gamma (P.1) | VOC | Positive |
| | SARS-CoV-2 | NCCP-43381 | alpha (B.1.1.7) | VOC | Positive |
| | SARS-CoV-2 | NCCP-43382 | beta (B.1.351) | VOC | Positive |
| | SARS-CoV-2 | NCCP-43390 | delta (B.1.617.2) | VOC | Positive |
| | SARS-CoV-2 | NCCP-43384 | epsilon (B.1.427) | VOI | Positive |
| | SARS-CoV-2 | NCCP-43385 | epsilon (B.1.429) | VOI | Positive |
| | SARS-CoV-2 | NCCP-43386 | eta (B.1.525) | VOI | Positive |
| | SARS-CoV-2 | NCCP-43387 | iota (B.1.526) | VOI | Positive |
| | SARS-CoV-2 | NCCP-43389 | kappa (B.1.617.1) | VOI | Positive |
| | SARS-CoV-2 | NCCP-43383 | zeta (P.2) | Not classified | Positive |
| | SARS-CoV-2 | NCCP-43330 | Not provided | Not classified | Positive |
| | SARS-CoV-2 | NCCP-43331 | Not provided | Not classified | Positive |
| | SARS-CoV-2 | NCCP-43342 | Not provided | Not classified | Positive |
| | SARS-CoV-2 | NCCP-43343 | Not provided | Not classified | Positive |
| | SARS-CoV-2 | NCCP-43344 | Not provided | Not classified | Positive |
| | SARS-CoV-2 | NCCP-43345 | Not provided | Not classified | Positive |
| | SARS-CoV-2 | Zeptometrix-0810587CFHI | Not provided | Not classified | Positive |
| | SARS-CoV-2 | Zeptometrix-0810589CFHI | Not provided | Not classified | Positive |
| | SARS-CoV-2 | Zeptometrix-0810590CFHI | Not provided | Not classified | Positive |
| | Coronavirus OC43 | ATCC VR1558 | Not provided | Not classification | Negative |
| | Coronavirus 229E | ATCC-VR 740 | Not provided | Not classification | Negative |
| | Influenza A virus | ATCC VR-810 | Not provided | Not classification | Negative |
| | Influenza B virus | ATCC VR-1735 | Not provided | Not classification | Negative |
| | Influenza A H1N1 | ATCC VR-1683 | Not provided | Not classification | Negative |
| | Influenza A H3N2 | ATCC VR-822 | Not provided | Not classification | Negative |
| | Influenza A H1N1 | ATCC VR-219 | Not provided | Not classification | Negative |
| | Influenza A H3N2 | ATCC VR-547 | Not provided | Not classification | Negative |
| | Respiratory syncytial virus A | ATCC VR-26 | Not provided | Not classification | Negative |
| | Respiratory syncytial virus B | ATCC VR-955 | Not provided | Not classification | Negative |
| | Parainfluenza type 1 | ATCC VR-1380 | Not provided | Not classification | Negative |
| Bacteria | *Staphylococcus aureus* | ATCC-29213 | Not provided | Not classification | Negative |
| | *Streptococcus pneumoniae* | ATCC-49619 | Not provided | Not classification | Negative |
| | *Streptococcus pyogenes* | ATCC-19615 | Not provided | Not classification | Negative |
| | *Pseudomonas aeruginosa* | ATCC-27853 | Not provided | Not classification | Negative |
| | *Enterobacter aerogenes* | ATCC-13048 | Not provided | Not classification | Negative |
| | *Enterobacter cloacae* | ATCC-13047 | Not provided | Not classification | Negative |
| | *Corynebacterium spp.* | ATCC-51860 | Not provided | Not classification | Negative |
| | *Moraxella catarrhalis* | KCCM-42706 | Not provided | Not classification | Negative |
| | *Haemophilus influenzae* | ATCC-9007 | Not provided | Not classification | Negative |
| Fungi | *Aspergillus fumigatus* | Zeptometrix-Z014 | Not provided | Not classification | Negative |
| | *Aspergillus flavus* | Zeptometrix-Z013 | Not provided | Not classification | Negative |
| | *Aspergillus niger* | Zeptometrix-Z105 | Not provided | Not classification | Negative |
| | *Aspergillus terreus* | Zeptometrix-Z016 | Not provided | Not classification | Negative |
| | *Aspergillus nidulans* | ATCC-38163 | Not provided | Not classification | Negative |
| | *Aspergillus versicolor* | ATCC-11730 | Not provided | Not classification | Negative |

Strains selected for assay validation (22 strains of SARS-CoV-2, 11 strains of other virus, 9 strains of bacteria, and 6 strains of fungi). Strain information, provided by the company from which the strain was acquired, is shown. Abbreviations: SARS-CoV-2, Severe acute respiratory syndrome-related coronavirus 2; Twistbio, Twist Bioscience; NCCP, National Culture Collection for Pathogens; ATCC, American Type Culture Collection; KCCM; Korean Culture Center of Microorganisms; VOC, variants of concern; VOI, variants of interest; CDC, Centers for Disease Control and Prevention.

## Analytical performance of the 5-SSG primer assay

The comparative limit of detection (LOD) of the 5-SSG primer assay was determined using heat-inactivated cultural fluids of SARS-CoV-2 (Zeptometrix-0810589CFHI) as a positive control, following the manufacturer's instructions. Each control was 10-fold serially diluted to approximately $1.4 \times 10^3$, $1.4 \times 10^2$, $1.4 \times 10^1$, $1.4 \times 10^0$, $1.4 \times 10^{-1}$, and $1.4 \times 10^{-2}$ $TCID_{50}$ (median tissue culture infectious dose)/mL. For the performance analysis of 5-SSG primers, 25 replicates were performed. The comparative LOD was determined as the minimum detectable concentration. Probit regression was used to estimate positive values with 95% confidence intervals [24].

## One-step RT-PCR and agarose gel electrophoresis

The template (2.5 ng) from viral, bacterial, and fungal strains was added for one-step RT-PCR (Nanohelix Co., Daejeon, Republic of Korea) analysis, which was performed using a SeeAmp (Seegene) instrument. PCR assays with the 5-SSG primers were performed using the following thermal cycling conditions: 45˚C for 15 min (reverse transcription), followed by 94˚C for 15 min (initial denaturation), and 45 cycles of 94˚C for 10 s (denaturation), 64˚C for 30 s (annealing), and 72˚C for 30 s (extension). A final extension step was conducted at 72˚C for 5 min. Next, the PCR products were analyzed using 2% agarose gel electrophoresis with 0.5× TBE buffer, and the gels were stained with ethidium bromide (Biosesang, Seongnam, Republic of Korea). PCR amplicons from the 67 samples were analyzed using agarose gel electrophoresis in a horizontal unit (CBS Scientific, San Diego, CA, USA) operating at 280 V for 28 min, and the band sizes on ethidium bromide-stained gels were quantified using a Gel-Doc XR+ system (Bio-Rad Laboratories, Hercules, CA, USA).

## PCR product purification and sequence analysis

All PCR-positive products were purified with MEGAquick-spin™ plus (iNtRON Biotechnology, Seongnam, Republic of Korea), according to the manufacturer's instructions [25]. The sequence analysis of PCR products (partial S gene amplified to ~800 bp) was performed using the 5-SSG primers (5′ tagged M13 primer) and the BigDye Terminator v3.1 cycle sequencing kit reagent (Applied Biosystems, Foster City, CA, USA). The sequence analysis conditions were as follows: 96˚C for 1 min (incubation), followed by 25 cycles of 96˚C for 10 s (denaturation), 50˚C for 5 s (annealing), and 60˚C for 4 min (extension). Dye-labeled products were analyzed using an ABI 3730 sequencer (Applied Biosystems). Sequencing chromatograms were analyzed manually using Variant Reporter™ v3.0 software (Applied Biosystems). Samples were classified as mutants if the sequencing results from the specific regions matched those of lineage information [26].

## NGS and data analysis

NGS was performed using the SARS-CoV-2 FLEX Panels (Paragon Genomics, Hayward, CA, USA) and an Illumina MiSeq platform (Illumina, San Diego, CA, USA) in accordance with the manufacturer's instructions [27]. Reverse transcription was performed using 55 ng of nucleic acid, and multiplex PCR was performed using 343 pairs of primers. A second PCR was conducted using CleanPlex Dual-Indexed PCR Primers for Illumina® Set A (Paragon Genomics). The final library was sequenced on an Illumina MiSeq platform (Illumina) with $2 \times 150$bp flow cells using a MiSeq Micro Reagent Kit v2 (300 cycles).

Next, NGS assays were analyzed using the Flomics pipeline (Flomics, Barcelona, Spain). The processing pipeline comprised FastQC v0.11.9 (quality control), followed by fastp v0.20.1.

(adapter trimming), Bowtie2 (reference alignment), and iVar v1.2.2. (variant calling). The viral lineage was accessed using the GISAID database, and PANGO Lineages [28]. In NGS analysis, depths of less than 10× were identified by read-depth segmentation in an integrated genomics viewer [27].

### Ethics statement

Ethical aspects of this study were reviewed and approved by the Seegene Medical Foundation Institutional Review Board (approval number, SMF-IRB-2021-006), provided that after conducting the laboratory diagnoses of SARS-CoV-2 testing, the remaining samples be destroyed. All data were fully anonymized administrative data without patient identifiers, and patient consent was waived by the institutional review board.

## Results

### Optimization of five SARS-CoV-2 primer pairs for S mutants

The 5-SSG primers consisted of five primer pairs, including 69S, 144S, 417S, 484S, and 570S. The 69S primer pair for 4 mutants (A67V, ΔH69/V70, D80A, and T95I), 144S primer pair for 9 mutants (D138Y, G142D, ΔY144, W152C, E154K, ΔE156/F157, R158G, R190S, and D215G), 417S primer pair for 4 mutants (D253G, K417T, K417N, and L452R), 484S primer pair for 4 mutants (T478K, E484K, E484Q, and N501Y), and 570S primer pair for 8 mutants (A570D, D614G, H655Y, Q677H, P681H, P681R, A701V, and T716I) were designed to detect target mutants (Table 1). The lineage and CDC classification information of each primer are shown in S1 Table. All target mutants were efficiently included in S gene coverage (Table 1, Fig 1, and S1 Table).

### PCR efficiency and 5-SSG primers performance analysis

The analytical performance of the 5-SSG primers was confirmed using a total of 67 strains, including viruses, bacteria, and fungi. The PCR results were determined to be positive or negative based on the expected PCR product sizes (Tables 1, 2 and S2). As shown in Tables 2 and S2, the 5-SSG

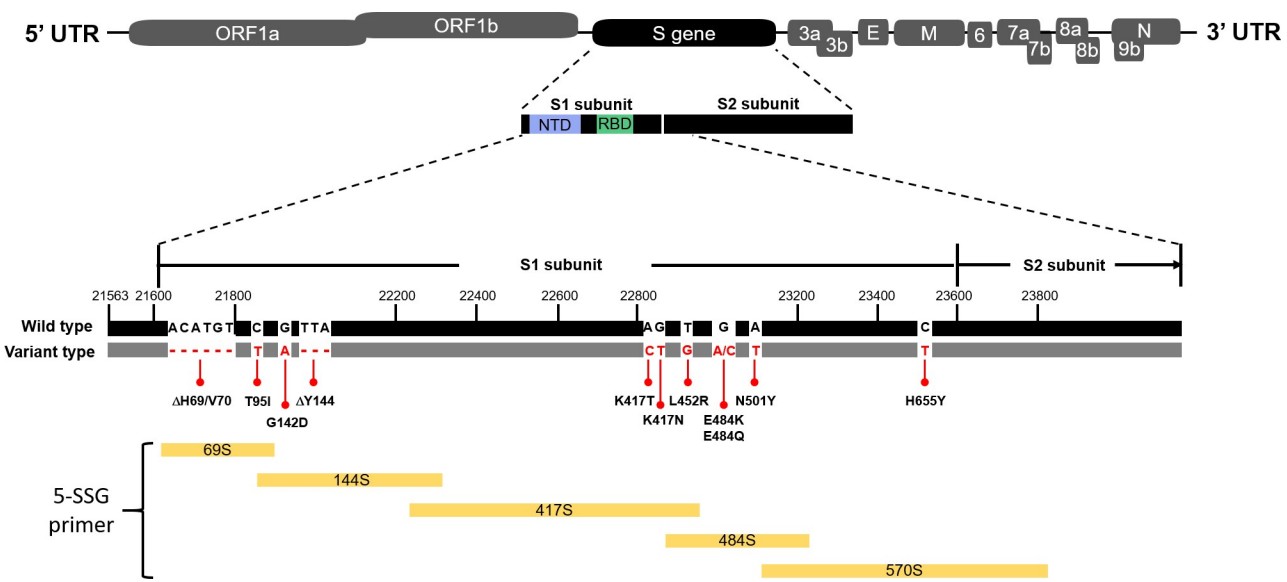

**Fig 1. Overall schematic structures of SARS-CoV-2 spike gene and derived 5-SSG primers.**

primer pairs achieved consistent results for twenty-two strains of SARS-CoV-2, whereas a negative result was obtained for the remaining 45 stains (other viruses, bacteria, and fungi).

## Determination of the analytical sensitivity of the 5-SSG primers

Analytical sensitivity, positivity rate, and LOD were estimated using 25 replicates of positive strains (Zeptometrix-0810590CFHI) at six different concentrations, from approximately $1.4 \times 10^{-2}$ to $1.4 \times 10^{3}$ TCID$_{50}$/mL (Table 3). Results (probit analysis) showed that the 95% LOD was approximately $3.7 \times 10^{1}$ TCID$_{50}$/mL for 69S, $9.8 \times 10^{1}$ TCID$_{50}$/mL for 144S, $6.6 \times 10^{1}$ TCID$_{50}$/mL for 417S, $3.9 \times 10^{1}$ TCID$_{50}$/mL for 484S, and $7.2 \times 10^{1}$ TCID$_{50}$/mL for 570S (Table 3). Assay results showed 100% reproducibility for all 5-SSG primer pairs, even for concentrations of as low as approximately $1.4 \times 10^{2}$ TCID$_{50}$/mL. The LOD was approximately $1.4 \times 10^{1}$ TCID$_{50}$/mL, except for in the 69S and 484S assays, which were 10 times more sensitive than the 144S, 417S, and 570S assays.

**Table 3. Evaluation of detection limit in target regions.**

| Primer pair | Conc. (TCID$_{50}$/mL) | Reactions | Positive | Positive rate (%) | LOD 95% level (TCID$_{50}$/mL) |
|---|---|---|---|---|---|
| 69S | $1.4 \times 10^{3}$ | 25 | 25 | 100 | $3.7 \times 10^{1}$ |
| | $1.4 \times 10^{2}$ | 25 | 25 | 100 | |
| | $1.4 \times 10^{1}$ | 25 | 20 | 80 | |
| | $1.4 \times 10^{0}$ | 25 | 7 | 28 | |
| | $1.4 \times 10^{-1}$ | 25 | 0 | 0 | |
| | $1.4 \times 10^{-2}$ | 25 | 0 | 0 | |
| 144S | $1.4 \times 10^{3}$ | 25 | 25 | 100 | $9.8 \times 10^{1}$ |
| | $1.4 \times 10^{2}$ | 25 | 25 | 100 | |
| | $1.4 \times 10^{1}$ | 25 | 4 | 16 | |
| | $1.4 \times 10^{0}$ | 25 | 0 | 0 | |
| | $1.4 \times 10^{-1}$ | 25 | 0 | 0 | |
| | $1.4 \times 10^{-2}$ | 25 | 0 | 0 | |
| 417S | $1.4 \times 10^{3}$ | 25 | 25 | 100 | $6.6 \times 10^{1}$ |
| | $1.4 \times 10^{2}$ | 25 | 25 | 100 | |
| | $1.4 \times 10^{1}$ | 25 | 8 | 32 | |
| | $1.4 \times 10^{0}$ | 25 | 0 | 0 | |
| | $1.4 \times 10^{-1}$ | 25 | 0 | 0 | |
| | $1.4 \times 10^{-2}$ | 25 | 0 | 0 | |
| 484S | $1.4 \times 10^{3}$ | 25 | 25 | 100 | $3.9 \times 10^{1}$ |
| | $1.4 \times 10^{2}$ | 25 | 25 | 100 | |
| | $1.4 \times 10^{1}$ | 25 | 18 | 72 | |
| | $1.4 \times 10^{0}$ | 25 | 1 | 4 | |
| | $1.4 \times 10^{-1}$ | 25 | 0 | 0 | |
| | $1.4 \times 10^{-2}$ | 25 | 0 | 0 | |
| 570S | $1.4 \times 10^{3}$ | 25 | 25 | 100 | $7.2 \times 10^{1}$ |
| | $1.4 \times 10^{2}$ | 25 | 25 | 100 | |
| | $1.4 \times 10^{1}$ | 25 | 7 | 28 | |
| | $1.4 \times 10^{0}$ | 25 | 0 | 0 | |
| | $1.4 \times 10^{-1}$ | 25 | 0 | 0 | |
| | $1.4 \times 10^{-2}$ | 25 | 0 | 0 | |

The 5-SSG primer-PCR reactions performed using ten-fold diluted positive samples. The LOD 95% data were estimated using the probit regression analysis.

Abbreviations: Conc., concentration; TCID$_{50}$, median tissue culture infective dose; LOD, limit of detection.

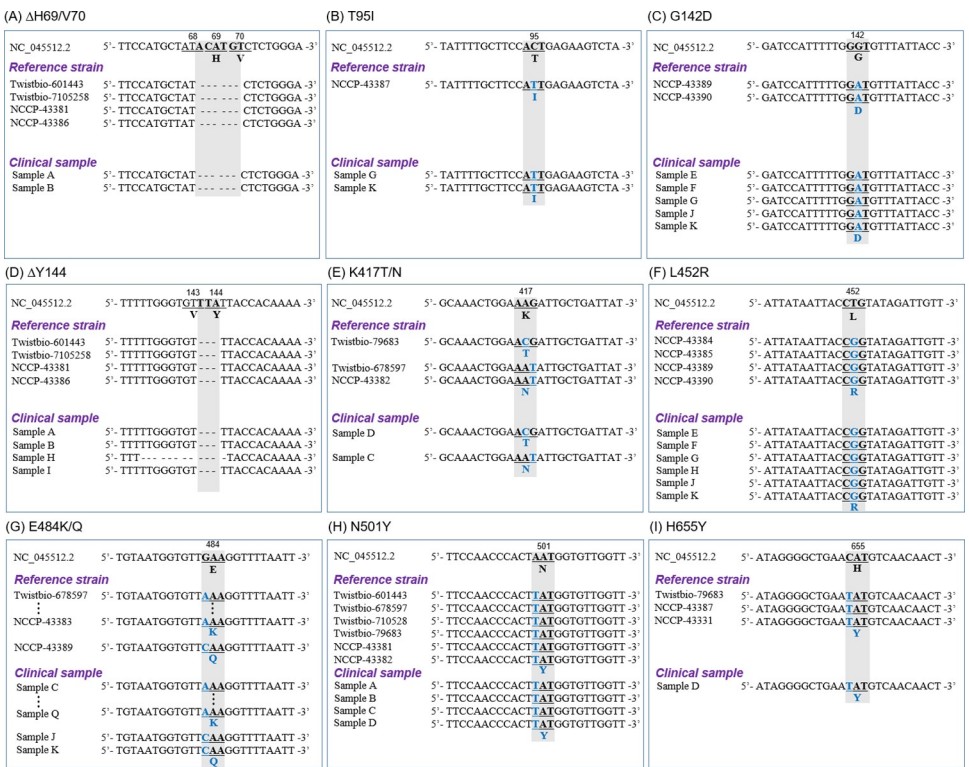

**Fig 2. Sequence analysis of SARS-CoV-2 S protein.** (A) ΔH69/V70, and (B) T95I from 69S; (C) G142D, and (D) ΔY144 from 144S; (E) K417T/N, and (F) L452R from 417S; (G) E484K/Q, and (H) N501Y from 484S; (I) H655Y from 570S. Sequences showing deletions or conversions are highlighted for comparison with the Wuhan-Hu-1-CoV sequence.

## Sanger sequencing analysis

As shown in Table 2, the nucleotide sequences of the positive PCR products obtained from 22 strains were compared with the existing S mutations through the Sanger Sequencing method using the M13 primer. The key deletion mutations ΔH69/V70 and ΔY144 were found in four strains (Twistbio-601443, Twistbio-7105258, NCCP-43381, and NCCP-43386). In addition, other substitution mutations were found to be 100% consistent with those in each strain's corresponding lineage, except for two cases in which substitutions at T95I for the NCCP-43390 strain and W152C for the NCCP-43384 strain were mismatched in the CDC classification (Figs 2 and S1, and S1 Table). Overall, it was confirmed through Sanger sequencing that the 5-SSG primers can detect predominant S gene mutations of SARS-CoV-2 observed in the major mutant strain categories, VOIs and VOCs, with high sensitivity and efficiency.

## Comparison of mutants detected by 5-SSG primer assay using Sanger sequencing versus NGS

SARS-CoV-2 mutants have been genetically characterized using NGS-based lineages [13,29]. To confirm the detection accuracy of the 5-SSG primer assay developed in this study for the SARS-CoV-2 mutants, NGS analysis results were used for a comparison. The NGS assay identified three strains (Twistbio-710528, Twistbio-601443, and NCCP-43381) as B.1.1.7, two (Twistbio-678597 and NCCP-43382) as B.1.351, one (Twistbio-79683) as P.1, one (NCCP-43390) as B.1.617.2, one (NCCP-43384) as B.1.427, one (NCCP-43385) as B.1.429, one (NCCP-43386) as B.1.525, one (NCCP-43387) as B.1.526, and one (NCCP-43389) as B.1.617.1.

**Table 4. Comparison of 5-SSG primers target mutations sequence of Sanger sequencing and NGS.**

| Grades of concern | Lineage | Source | Sequence analysis method | | Final Determination |
|---|---|---|---|---|---|
| | | | NGS | Sanger sequencing | |
| VOC | B.1.1.7 | Twistbio-710528 | ΔH69/V70, ΔY144, N501Y, A570D, D614G, P681H, T716I | ΔH69/V70, ΔY144, N501Y, A570D, D614G, P681H, T716I | Match |
| | | Twistbio-601443 | ΔH69/V70, ΔY144, N501Y, A570D, D614G, P681H, T716I | ΔH69/V70, ΔY144, N501Y, A570D, D614G, P681H, T716I | Match |
| | | NCCP-43381 | ΔH69/V70, ΔY144, N501Y, A570D, D614G, P681H, R682Q, T716I | ΔH69/V70, ΔY144, N501Y, A570D, D614G, P681H, R682Q, T716I | Match |
| | B.1.351 | Twistbio-678597 | D80A, D215G, ΔLAL242-244, K417N, E484K, N501Y, D614G, A701V | D80A, D215G, ΔLAL242-244, K417N, E484K, N501Y, D614G, A701V | Match |
| | | NCCP-43382 | L54F, D80A, D215G, ΔLAL242-244, K417N, E484K, N501Y, D614G, A701V | L54F, D80A, D215G, ΔLAL242-244, K417N, E484K, N501Y, D614G, A701V | Match |
| | P.1 | Twistbio-79683 | D138Y, R190S, K417T, E484K, N501Y, D614G, H655Y | D138Y, R190S, K417T, E484K, N501Y, D614G, H655Y | Match |
| | B.1.617.2 | NCCP-43390 | G142D, ΔE156/F157, R158G, L452R, T478K, Q613H, D614G, P681R, R682W | G142D, ΔE156/F157, R158G, L452R, T478K, Q613H, D614G, P681R, R682W | Match |
| VOI | B.1.427 | NCCP-43384 | W152C, L452R, D614G | W152C, L452R, D614G | Match |
| | B.1.429 | NCCP-43385 | W152C, L452R, D614G | W152C, L452R, D614G | Match |
| | B.1.525 | NCCP-43386 | Q52R, A67V, ΔH69/V70, ΔY144, E484K, D614G, Q677H | Q52R, A67V, ΔH69/V70, ΔY144, E484K, D614G, Q677H | Match |
| | B.1.526 | NCCP-43387 | T95I, D253G, E484K, D614G, H655Y, A701V | T95I, D253G, E484K, D614G, H655Y, A701V | Match |
| | B.1.617.1 | NCCP-43389 | G142D, E154K, L452R, E484Q, D614G, P681R, R682Q | G142D, E154K, L452R, E484Q, D614G, P681R, R682Q | Match |
| Not included | P.2 | NCCP-43383 | E484K, D614G | E484K, D614G | Match |
| | B | NCCP-43330 | - | - | Match |
| | A | NCCP-43331 | H655Y | H655Y | Match |
| | B | NCCP-43342 | - | - | Match |
| | B.1.1- | NCCP-43343 | D614G, R682Q | D614G, R682Q | Match |
| | B.1- | NCCP-43344 | D215H, D614G, R682Q | D215H, D614G, R682Q | Match |
| | B.1.497 | NCCP-43345 | D614G, ΔQTQTN675-679, R682L | D614G, ΔQTQTN675-679, R682L | Match |
| | A | Zeptometrix-0810587CFHI | D215/L216insKLRS, ΔQTQTN675-679 | D215/L216insKLRS, ΔQTQTN675-679 | Match |
| | B | Zeptometrix-0810589CFHI | N74K, S247R, ΔNSPRRARSVA679-688 | N74K, S247R, ΔNSPRRARSVA679-688 | Match |
| | A | Zeptometrix-0810590CFHI | S247R, V367F, R682Q, | S247R, V367F, R682Q | Match |

Abbreviations: VOC, Variant of concern; VOI, variant of interest; Δ, deletion. Low-coverage NGS data are marked in underline.

The remaining ten strains (NCCP-43330, NCCP-43331, NCCP-43342, NCCP-43343, NCCP-43344, NCCP-43345, NCCP-43383, Zeptometrix-0810587CFHI, Zeptometrix-0810589CFHI, and Zeptometrix-0810589CFHI) were genetically classified into another lineage (Table 4). Results of NGS and Sanger sequencing using the 5-SSG primers showed 100% consistency for all strains, including T95I for the NCCP-43390 strain and W152C for the NCCP-43384 strain (Table 4). Taken together, the 5-SSG primer assay is very efficient in detecting SARS-CoV-2 major S gene mutant strains.

## Validation of clinical sample variants using the 5-SSG primers

To confirm the detection accuracy of the 5-SSG primer assay using clinical samples, Sanger sequencing results were compared with those of NGS analysis (Table 5). The results of VOCs

**Table 5. Validation of 5-SSG primers target mutations sequence using clinical samples.**

| Grades of concern | Lineage | Sample | Sequence analysis method | | Final Determination |
|---|---|---|---|---|---|
| | | | NGS | Sanger sequencing | |
| VOC | B.1.1.7 | Sample A | ΔH69/V70, ΔY144, N501Y, A570D, D614G, P681H, T716I | ΔH69/V70, ΔY144, N501Y, A570D, D614G, P681H, T716I | Match |
| | | Sample B | ΔH69/V70, ΔY144, N501Y, A570D, D614G, P681H, T716I | ΔH69/V70, ΔY144, N501Y, A570D, D614G, P681H, T716I | Match |
| | B.1.351 | Sample C | D80A, D215G, ΔLAL242-244, K417N, E484K, N501Y, D614G, A701V | D80A, D215G, ΔLAL242-244, K417N, E484K, N501Y, D614G, A701V | Match |
| | P.1 | Sample D | D138Y, R190S, K417T, E484K, N501Y, D614G, H655Y | D138Y, R190S, K417T, E484K, N501Y, D614G, H655Y | Match |
| | B.1.617.2 | Sample E | G142D, ΔE156/F157, R158G, L452R, T478K, D614G, P681R | G142D, ΔE156/F157, R158G, L452R, T478K, D614G, P681R | Match |
| | | Sample F | G142D, ΔE156/F157, R158G, L452R, T478K, D614G, P681R | G142D, ΔE156/F157, R158G, L452R, T478K, D614G, P681R | Match |
| | | Sample G | T95I, G142D, ΔE156/F157, R158G, L452R, T478K, D614G, P681R | T95I, G142D, ΔE156/F157, R158G, L452R, T478K, D614G, P681R | Match |
| VOI | B.1.429 | Sample H | ΔLGVY141-144, W152C, G252V, S256L, L452R, D614G | ΔLGVY141-144, W152C, G252V, S256L, L452R, D614G | Match |
| | B.1.525 | Sample I | Q52R, A67V, ΔH69/V70, ΔY144, E484K, D614G, Q677H | Q52R, A67V, ΔH69/V70, ΔY144, E484K, D614G, Q677H | Match |
| | B.1.617.1 | Sample J | G142D, E154K, L452R, E484Q, D614G, P681R | G142D, E154K, L452R, E484Q, D614G, P681R | Match |
| | | Sample K | T95I, G142D, E154K, L452R, E484Q, D614G, P681R | T95I, G142D, E154K, L452R, E484Q, D614G, P681R | Match |
| Not included | B.1.497 | Sample L | D614G | D614G | Match |
| | | Sample M | D614G | D614G | Match |
| | | Sample N | D614G | D614G | Match |
| | B.1.619 | Sample O | I210T, N440K, E484K, D614G | I210T, N440K, E484K, D614G | Match |
| | | Sample P | I210T, N440K, E484K, D614G | I210T, N440K, E484K, D614G | Match |
| | | Sample Q | I210T, N440K, E484K, D614G | I210T, N440K, E484K, D614G | Match |

Abbreviations: VOC, Variant of concern; VOI, variant of interest; Δ, deletion. Low-coverage NGS data are marked in underline.

(B.1.1.7, B.1.351, P.1, and B.1.617.2), VOIs (B.1.429, B.1.525, and B.1.617.1), and the remaining two lineages (B.1.497, B.1.619) were compared (Table 5). NGS assays and Sanger sequencing using the 5-SSG primers showed 100% consistent results for all strains. We concluded that the 5-SSG primer assay also had a very efficient performance with clinical samples.

## Discussion

In the ongoing COVID-19 pandemic, it has been demonstrated that the rapid detection of the pathogen is critical to prevent the rampant spread of the disease [30]. The emergence of SARS-CoV-2 variants, which are associated with increased transmission, disease severity, and resistance to vaccines, is a grave concern [31]. The alpha (B.1.1.7) and beta (B.1.351) lineages of SARS-CoV-2, which account for 98.7% of total variant cases, contain the mutations ΔH69/V70, E484K, and N501Y [32]. S protein-based vaccines might provide less protection against these mutants (ΔH69/V70, E484K, and N501Y) of SARS-CoV-2 [33]. Therefore, a simple and

rapid screening assay to monitor the emergence and spread of these variants is essential to implement public health strategies [31].

In this study, we developed primers for the rapid and accurate detection of the key mutants of the S gene of SARS-CoV-2 and evaluated the reliability and reproducibility of these primers (Tables 2 and 3). The 5-SSG primers (69S, 144S, 417S, 484S, and 570S) had high analytical specificity for SARS-CoV-2 strains and no cross-reactivity with other strains (Tables 2 and S2). Results of Sanger sequencing using 5-SSG primers and commercial NGS were in 100% agreement; however, the three approaches differed in their ability to detect the E484K and D215G variants of the beta (B.1.351) lineage, E484K of the gamma (P.1) lineage, and G142D of the delta (B.1.617.2) lineage (Tables 4 and 5). These results indicate that in NGS analysis, low-depth levels of mutants (G142D, D215G, and E484K) are detected, because the target amplification is affected by a mutation in the reverse primer binding site (ΔE156/F157, R158G, ΔLAL242-244, and N501Y). In addition, NGS is limited to the environment in which the equipment is built, and it also takes a longer as it is more complex than typical Sanger sequencing [34]. Therefore, the Sanger sequencing-based 5-SSG primer assay system can rapidly and accurately detect key mutants of the S gene without resource constraint, and is a useful tool that can overcome the limitation of relatively low read-depth caused by mutations in primer-binding site during NGS analysis.

One limitation of this study is that the performance of the 5-SSG primers was tested using small numbers of clinical samples through Sanger sequencing and NGS analysis, and thus, further studies using a larger number of clinical samples should be performed. In addition, the current 5-SSG primer system can identify lambda (C.37) variants with the 417S primer set, but the ΔRSYLTPGD246-253N mutation affects the 144S reverse primer. Therefore, improvements in primer performance for detection of additional variants (e.g. B. 1.617.3 and B. 1.621) and the development of new primers should be pursued in future studies.

Collectively, the 5-SSG primer assay system has high PCR sensitivity specifically for SARS-CoV-2 and is a useful tool that can detect various S gene mutants very quickly and accurately, thereby contributing to the faster control of pathogen transmission in the population.

## Supporting information

**S1 Fig. Sequence chromatograms of raw data for SARS-CoV-2 S protein.** (A) ΔH69/V70 and (B) T95I from 69S; (C) G142D and (D) ΔY144 from 144S; (E) K417T, (F) K417N, and (G) L452R from 417S; (H) E484K, (I) E484Q, and (J) N501Y from 484S; (K) H655Y from 570S. Chromatograms showing deletions or conversions are highlighted for comparison with the Wuhan-Hu-1-CoV sequence.
(TIF)

**S1 Table. Primers-specific target mutant and lineage classification, and Sanger sequencing result.** Abbreviations: VOI, Variants of Interest; VOC, Variants of Concern; Twistbio, Twist Bioscience; NCCP, National Culture Collection for Pathogens.
(PDF)

**S2 Table. PCR results and non-respiratory pathogen strain information used in this study.** Abbreviations: SARS-CoV-2, Severe acute respiratory syndrome-related coronavirus 2; ATCC, American Type Culture Collection; KBPV, Korea Bank for Pathogen Viruses; NIBSC, National Institute for Biological Standards and Control; KCTC, Korean Collection for Type Cultures.
(PDF)

**S1 Raw data.**
(ZIP)

## Acknowledgments

We would like to thank Editage for English language editing. We also thank the National Culture Collection for Pathogens for kindly providing fifteen strains of SARS-CoV-2 as resources (NCCP-43330, NCCP-43331, NCCP-43342, NCCP-43343, NCCP-43344, NCCP-43345, NCCP-43381, NCCP-43382, NCCP-43383, NCCP-43384, NCCP-43385, NCCP-43386, NCCP-43387, NCCP-43389, and NCCP-43390) in this study.

## Author Contributions

**Conceptualization:** Jung Eun Park, Yong-Jin Yang.

**Data curation:** Hye Soo Jung, Youngjin Kwon, Inhee Kim, Dong Kwan Kim.

**Formal analysis:** Ho Jae Lim, Hye Soo Jung, Youngjin Kwon, Nae Yu.

**Investigation:** Min Young Park, Nackmoon Sung, Sun-Hwa Lee.

**Methodology:** Ho Jae Lim, Min Young Park, Jung Eun Park, Yong-Jin Yang.

**Supervision:** Min Young Park, Yong-Jin Yang.

**Validation:** Ho Jae Lim, Min Young Park, Jung Eun Park, Yong-Jin Yang.

**Writing – original draft:** Ho Jae Lim, Jung Eun Park.

**Writing – review & editing:** Ho Jae Lim, Min Young Park, Nae Yu, Nackmoon Sung, Sun-Hwa Lee, Yong-Jin Yang.

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
