## [Decision Letter · Decision Letter 0]

20 Sep 2021

PONE-D-21-25453Development of efficient Sanger sequencing-based assay system for SARS-CoV-2 spike variantsPLOS ONE

Dear Dr. Yang,

Thank you for submitting your manuscript to PLOS ONE. After careful consideration, we feel that it has merit but does not fully meet PLOS ONE’s publication criteria as it currently stands. Therefore, we invite you to submit a revised version of the manuscript that addresses the points raised during the review process.

I have received the reviews of your manuscript. While your paper addresses an interesting question, the reviewers stated several concerns about your study and did not recommend publication in its present form.  The presentation as well as the readability of the manuscript need to be improved, since the manuscript as a whole is quite convoluted.  The title of the manuscript implied that the PCR plus Sanger Sequencing provided the necessary information.  However, this is not clearly conveyed in the main text.  The abstract only mentioned the 5-SSG primer assay development.  From the main text, the 5-SSG primer assay system does not include sequence, how do the authors determine the mutation detected, PCR product size?  In addition, the quality of the language needs to be improved, there are quite a few awkward sentences and typos throughout the manuscript.  Please have a fluent, preferably native, English-language speaker thoroughly copyedit your manuscript for language usage, spelling, and grammar.  

Specific comments:

Need to declare that Jeong-Eui Lee is the Seegene employeeLine 88 – 91, please rephrase for clarity. What really did the authors develop?  PCR assay in conjunction with Sanger sequencing or just PCR assay?Line 99, “redesigned or modified” from what?Line 117 – 120, awkward sentence, please rephrase for clarity. Also, are these viruses, bacteria and fungi selected causing respiratory infections? What is the selection criteria?Line 119, suggest changing “2 other kinds of coronaviruses” to “Coronavirus OC43 & 229E”Line 135 – 137, awkward sentences, please rephrase for clarity.Line 146 – 147, this sentence is confusing, please rephrase.Line 198 – 199, please rephrase for clarity.

We look forward to receiving your revised manuscript.

Kind regards,

Baochuan Lin, Ph.D.

Academic Editor

PLOS ONE

“No”

3. Thank you for providing the following Funding Statement: 

“No”

We note that one or more of the authors is affiliated with the funding organization, indicating the funder may have had some role in the design, data collection, analysis or preparation of your manuscript for publication; in other words, the funder played an indirect role through the participation of the co-authors.

If the funding organization did not play a role in the study design, data collection and analysis, decision to publish, or preparation of the manuscript and only provided financial support in the form of authors' salaries and/or research materials, please review your statements relating to the author contributions, and ensure you have specifically and accurately indicated the role(s) that these authors had in your study in the Author Contributions section of the online submission form. Please make any necessary amendments directly within this section of the online submission form.  Please also update your Funding Statement to include the following statement: “The funder provided support in the form of salaries for authors [insert relevant initials], but did not have any additional role in the study design, data collection and analysis, decision to publish, or preparation of the manuscript. The specific roles of these authors are articulated in the ‘author contributions’ section.”

If the funding organization did have an additional role, please state and explain that role within your Funding Statement.

Please also provide an updated Competing Interests Statement declaring this commercial affiliation along with any other relevant declarations relating to employment, consultancy, patents, products in development, or marketed products, etc. 

Reviewers' comments:

Reviewer's Responses to Questions

**Comments to the Author**

1. Is the manuscript technically sound, and do the data support the conclusions?

Reviewer #1: Yes

Reviewer #2: Yes

2. Has the statistical analysis been performed appropriately and rigorously? 

Reviewer #1: N/A

Reviewer #2: Yes

3. Have the authors made all data underlying the findings in their manuscript fully available?

Reviewer #1: No

Reviewer #2: Yes

4. Is the manuscript presented in an intelligible fashion and written in standard English?

Reviewer #1: Yes

Reviewer #2: Yes

5. Review Comments to the Author

Reviewer #1: Lim et al. have developed SARS-CoV-2 Spike protein primer sets to distinguish VOC and VOI as defined by WHO. The primers seem to work well for the VOC strains and some of the VOI strains. However, recently, lambda (June 14) and mu (August 30) have been newly designated. Since mu is designated after the submission date of August 6, authors can dismiss the VOI. The lambda (C.37) has 246_253delinsN deletion, which coincides with the reverse primer of 144S. Authors need to check whether the current primer design can detect the deletion in the lambda variant. Also, authors needs to share the raw data of the sequencing data prior to the publication.

Reviewer #2: This study describes development and testing of five primer sets for detecting SARS-CoV-2 spike variants using Sanger sequencing. The manuscript is well written, and was a pleasure to read. The study appears robust, apt, and timely, especially considering its potential for application in clinical settings.

I only have a few minor comments:

1. A few typos:

-line 38, 'detect nine', instead of 'detect of nine'

-line 43, 'ability of the 5-SSG primer', instead of 'ability of 5-SSG primer'. Same for lines 250, 273, 284.

-line 81, 'methods', instead of 'method'

2. In the first paragraph of the results section (lines 204-214) and throughout the manuscript, the authors refer to 'mutants' as 'variants'. The term 'variant' in SARS-CoV-2 literature is probably more appropriately used for a constellation of mutations that make up a genetically (and usually epidemiologically) distint virus, rather than single mutations, and I fear that the authors use of the word here may not be appropriate.

3. Under acknowledgement, the name of the person being 'thanked' is missing.

6. PLOS authors have the option to publish the peer review history of their article (what does this mean?). If published, this will include your full peer review and any attached files.

Reviewer #1: **Yes: **Takahiko Koyama

Reviewer #2: No

---

## [Author Response · Author response to Decision Letter 0]

13 Oct 2021

Responses to Academic Editor Comments

Thank you for submitting your manuscript to PLOS ONE. After careful consideration, we feel that it has merit but does not fully meet PLOS ONE’s publication criteria as it currently stands. Therefore, we invite you to submit a revised version of the manuscript that addresses the points raised during the review process.

I have received the reviews of your manuscript. While your paper addresses an interesting question, the reviewers stated several concerns about your study and did not recommend publication in its present form. The presentation as well as the readability of the manuscript need to be improved, since the manuscript as a whole is quite convoluted. The title of the manuscript implied that the PCR plus Sanger Sequencing provided the necessary information. However, this is not clearly conveyed in the main text. The abstract only mentioned the 5-SSG primer assay development. From the main text, the 5-SSG primer assay system does not include sequence, how do the authors determine the mutation detected, PCR product size? In addition, the quality of the language needs to be improved, there are quite a few awkward sentences and typos throughout the manuscript. Please have a fluent, preferably native, English-language speaker thoroughly copyedit your manuscript for language usage, spelling, and grammar.

A: Thank you for your valuable comment. To improve the quality of the manuscript's expression and readability, we requested a native-speaker proofreading service, and the revised manuscript has been resubmitted here. Also, as per your suggestion, the title has been changed to “Development of an efficient Sanger sequencing-based assay for detecting SARS-CoV-2 spike mutations", to more accurately include the essential information provided by PCR and Sanger sequencing. All additional corrections made in the manuscript are marked in red. Electrophoresis was performed to determine the size of the PCR product, and the results are summarized in Tables 2 and S2. For reference, photos of the electrophoresis have been attached below.

 

Specific comments:

1. Need to declare that Jeong-Eui Lee is the Seegene employee

A: We have carefully reconsidered the issue of Jeong-Eui Lee as a co-author. Since his direct contribution to the paper is not entirely clear, we have decided to remove his authorship. We apologize for any confusion we may have caused by not initially making a more careful decision.

2. Line 88 – 91, please rephrase for clarity. What really did the authors develop? PCR assay in conjunction with Sanger sequencing or just PCR assay?

A: We have modified the manuscript’s passage to say, “PCR assay in conjunction with Sanger sequencing”, and marked these changes in red.

3. Line 99, “redesigned or modified” from what?

A: We have modified the sentence passage to state, ““modified from Global Initiative on Sharing Avian Influenza Data (GISAID) database, with a frequency cut-off > 1%, applied with degenerative or inosine to optimize melting temperature (Tm), avoid repetitive sequences, and include GC content > 65%, using Gene Runner (ver. 6.0) [20, 21].”, and marked these changes in red. 

4. e 117 – 120, awkward sentence, please rephrase for clarity. Also, are these viruses, bacteria and fungi selected causing respiratory infections? What is the selection criteria?

 A: We have chosen here to focus on all pathogen species for which standard stains could be obtained to compare analytical capabilities. Among them, the results were separated between those causing respiratory infection (Table 2), and other pathogens (Supplementary Table 2) for clarity.

5. Line 119, suggest changing “2 other kinds of coronaviruses” to “Coronavirus OC43 & 229E”

A: As per your suggestion, we have changed the sentence to “Coronavirus OC43 & 229E”, and marked the revision in red.

6. Line 135 – 137, awkward sentences, please rephrase for clarity.

A: As follow your suggestion, we have reworded the sentences for clarity, and marked these changes in red.

7. Line 146 – 147, this sentence is confusing, please rephrase.

A: The sentence has been rephrased, and is marked in red.

8. Line 198 – 199, please rephrase for clarity.

A: The passage has been rephrased, and is marked in red. 

Responses to Reviewer Comments

Reviewer #1’s comments and responses

▶ Comment 1: Lim et al. have developed SARS-CoV-2 Spike protein primer sets to distinguish VOC and VOI as defined by WHO. The primers seem to work well for the VOC strains and some of the VOI strains. However, recently, lambda (June 14) and mu (August 30) have been newly designated. Since mu is designated after the submission date of August 6, authors can dismiss the VOI. The lambda (C.37) has 246_253delinsN deletion, which coincides with the reverse primer of 144S. Authors need to check whether the current primer design can detect the deletion in the lambda variant. Also, authors needs to share the raw data of the sequencing data prior to the publication.

▶ Response to comment 1: Thank you for your very important comments. We have checked whether the 5-SSG primer assay system can detect deletion in Mu and lambda variants. As you point out, the lambda (C.37) 246_253delinsN deletion is not amplified under the influence of 144S reverse primer, but it can be confirmed with the 417S primer set. In addition, there is no problem in confirming Mu (B. 1.621) mutations.

We have attached the figure below to support this information.

Since your comments are very important, so we will monitor the occurrence of mutations and continue to improve our primer sets. As your request, we have included a new supplementary figure of raw data for the sequencing results of the key mutants, and have enclosed a data file to share the NGS raw data in FASTA format.

Reviewer #2’s comments and responses

This study describes development and testing of five primer sets for detecting SARS-CoV-2 spike variants using Sanger sequencing. The manuscript is well written, and was a pleasure to read. The study appears robust, apt, and timely, especially considering its potential for application in clinical settings.

I only have a few minor comments:

▶ Comment 1: A few typos: 

- line 38, 'detect nine', instead of 'detect of nine'

- line 43, 'ability of the 5-SSG primer', instead of 'ability of 5-SSG primer'. Same for lines 250, 273, 284

- line 81, 'methods', instead of 'method' be appropriate.

▶ Response to comment 1: All errors in manuscript have been corrected, and are marked in red.

▶ Comment 2: In the first paragraph of the results section (lines 204-214) and throughout the manuscript, the authors refer to 'mutants' as 'variants'. The term 'variant' in SARS-CoV-2 literature is probably more appropriately used for a constellation of mutations that make up a genetically (and usually epidemiologically) distint virus, rather than single mutations, and I fear that the authors use of the word here may not be appropriate.

▶ Response to comment 2: We agree with your valuable comment. The inappropriate words in manuscript have been corrected and are marked in red.

▶ Comment 3: Under acknowledgement, the name of the person being 'thanked' is missing.

▶ Response to comment 3: We have been added the name of Editage in acknowledgement and marked it in red.

---

## [Decision Letter · Decision Letter 1]

10 Nov 2021

PONE-D-21-25453R1Development of an efficient Sanger sequencing-based assay for detecting SARS-CoV-2 spike mutationsPLOS ONE

Dear Dr. Yang,

Thank you for submitting your manuscript to PLOS ONE. After careful consideration, we feel that it has merit but does not fully meet PLOS ONE’s publication criteria as it currently stands. Therefore, we invite you to submit a revised version of the manuscript that addresses the points raised during the review process.

Both reviewers agreed that the revised version has addressed most of the comments raised and showed significant improvement.  However, one of the reviewers still feel that one significant point regarding delta variant needs to be addressed.  In addition, I also have a few points that still need to be addressed (see specific comments below). Specific comments:1. Line 135 - 136,  "Anonymized residual of 17 nasopharyngeal swab specimens SARS-CoV-2

positive..." suggest changing to "Anonymized residual of 17 SARS-CoV-2 positive nasopharyngeal swab specimens..."2. Line 146 - 147, "The comparative LOD assessment was performed 25 times until the concentration of the targeted band was detected as positive." Not sure what the authors want to convey, please rephrase for clarity.3. Line 150 & 152, change PCR to RT-PCR.

We look forward to receiving your revised manuscript.

Kind regards,

Baochuan Lin, Ph.D.

Academic Editor

PLOS ONE

Journal Requirements:

Reviewers' comments:

Reviewer's Responses to Questions

**Comments to the Author**

1. If the authors have adequately addressed your comments raised in a previous round of review and you feel that this manuscript is now acceptable for publication, you may indicate that here to bypass the “Comments to the Author” section, enter your conflict of interest statement in the “Confidential to Editor” section, and submit your "Accept" recommendation.

Reviewer #1: All comments have been addressed

Reviewer #2: All comments have been addressed

2. Is the manuscript technically sound, and do the data support the conclusions?

Reviewer #1: Yes

Reviewer #2: Yes

3. Has the statistical analysis been performed appropriately and rigorously? 

Reviewer #1: N/A

Reviewer #2: (No Response)

4. Have the authors made all data underlying the findings in their manuscript fully available?

Reviewer #1: Yes

Reviewer #2: Yes

5. Is the manuscript presented in an intelligible fashion and written in standard English?

Reviewer #1: Yes

Reviewer #2: Yes

6. Review Comments to the Author

Reviewer #1: Authors have responded to my review comments on 246_253delinsN on lambda variant. However, since this is an important point, authors should address that their assay works for the lambda and others in discussion.

Reviewer #2: Thank you for addressing my previous comments and others'. I think the current version of the manusript is much improved.

7. PLOS authors have the option to publish the peer review history of their article (what does this mean?). If published, this will include your full peer review and any attached files.

Reviewer #1: **Yes: **Takahiko Koyama

Reviewer #2: No

---

## [Author Response · Author response to Decision Letter 1]

15 Nov 2021

Specific comments

1. Line 135 - 136, "Anonymized residual of 17 nasopharyngeal swab specimens SARS-CoV-2 positive..." suggest changing to "Anonymized residual of 17 SARS-CoV-2 positive nasopharyngeal swab specimens..."

A: Thank you for your valuable comment. We have modified this text in the manuscript as “anonymized residual of 17 SARS-CoV-2 positive nasopharyngeal swab specimens of patients diagnosed with SARS-CoV-2 positive between February and June 2021 were obtained and used for this study” (lines 145-148).

2. Line 146 - 147, "The comparative LOD assessment was performed 25 times until the concentration of the targeted band was detected as positive." Not sure what the authors want to convey, please rephrase for clarity.

A: We have modified the sentence as follows for clarity: “For the performance analysis of 5-SSG primers, 25 replicates were performed. The comparative LOD was determined as the minimum detectable concentration” (lines 157-159). 

3. Line 150 & 152, change PCR to RT-PCR.

A: As per your suggestion, we have changed the subheading and sentence to RT-PCR (lines 162, 164).

 

Responses to Reviewer Comments

Reviewer #1’s comments and responses

▶ Comment: Authors have responded to my review comments on 246_253delinsN on lambda variant. However, since this is an important point, authors should address that their assay works for the lambda and others in discussion.

▶ Response to comment: Thank you for your valuable comment. As suggested, we mentioned the lambda variant issue in this study, as the 246_253delinsN mutation, in the discussion section of the manuscript (lines 337-341).

---

## [Editor Report · Decision Letter 2]

18 Nov 2021

Development of an efficient Sanger sequencing-based assay for detecting SARS-CoV-2 spike mutations

PONE-D-21-25453R2

Dear Dr. Yang,

We’re pleased to inform you that your manuscript has been judged scientifically suitable for publication and will be formally accepted for publication once it meets all outstanding technical requirements.

Kind regards,

Baochuan Lin, Ph.D.

Academic Editor

PLOS ONE
---

## [Editor Report · Acceptance letter]

6 Dec 2021

PONE-D-21-25453R2 

Development of an efficient Sanger sequencing-based assay for detecting SARS-CoV-2 spike mutations 

Dear Dr. Yang:

I'm pleased to inform you that your manuscript has been deemed suitable for publication in PLOS ONE. Congratulations! Your manuscript is now with our production department. 

Kind regards, 

on behalf of

Dr. Baochuan Lin 

Academic Editor

PLOS ONE